# Electrothermally Self-Healing Delamination Cracks in Carbon/Epoxy Composites Using Sandwich and Tough Carbon Nanotube/Copolymer Interleaves

**DOI:** 10.3390/polym14204313

**Published:** 2022-10-14

**Authors:** Qin Ouyang, Ling Liu, Zhanjun Wu

**Affiliations:** 1School of Aerospace Engineering & Applied Mechanics, Tongji University, Shanghai 200092, China; 2School of Aeronautics and Astronautics, Dalian University of Technology, Dalian 116024, China

**Keywords:** carbon nanotube, polymer matrix composites, toughening, self-healing, electrical heating

## Abstract

Herein, two sandwich and porous interleaves composed of carbon nanotube (CNT) and poly(ethylene-co-methacrylic acid) (EMAA) are proposed, which can simultaneously toughen and self-heal the interlaminar interface of a carbon fiber-reinforced plastic (CFRP) by in situ electrical heating of the CNTs. The critical strain energy release rate modes I (*G*_IC_) and II (*G*_IIC_) are measured to evaluate the toughening and self-healing efficiencies of the interleaves. The results show that compared to the baseline CFRP, the CNT-EMAA-CNT interleaf could increase the *G*_IC_ by 24.0% and the *G*_IIC_ by 15.2%, respectively, and their respective self-healing efficiencies could reach 109.7–123.5% and 90.6–91.2%; meanwhile, the EMAA-CNT-EMAA interleaf can improve the *G*_IC_ and *G*_IIC_ by 66.9% and 16.7%, respectively, and the corresponding self-healing efficiencies of the *G*_IC_ and *G*_IIC_ are 122.7–125.9% and 93.1–94.7%. Thus, both the interleaves show good toughening and self-healing efficiencies on the interlaminar fracture toughness. Specifically, the EMAA-CNT-EMAA interleaf possesses better multi-functionality, i.e., moderate toughening ability but notable self-healing efficiency via electrical heating, which is better than the traditional neat EMAA interleaf and oven-based heating healing method.

## 1. Introduction

Repairing matrix cracks, especially delamination cracks, is a major technical challenge and maintenance cost for carbon fiber-reinforced polymer matrix composites (CFRPs) in light load-bearing structural applications, such as aircraft, ships, automobiles, etc. [1,2,3]. Once delamination cracks coalesce and grow under cyclic loading, they will greatly reduce the stiffness and strength of structures, eventually leading to severe failure during their service life [4,5]. Although some promising one-off damage repair techniques have been proposed for CFRPs, the complexity and associated costs involved during the repair process greatly hinder the restoration of CFRP structural integrity [6,7]. Therefore, the development of self-healing CFRPs capable of resolving early damage formation has attracted great attention in recent years.

Self-healing CFRPs are a new category of composite materials that can autonomously repair their internal cracks via self-activation or external stimuli, thereby restoring their initial mechanical properties [8]. According to the incorporation method of self-healing agents, self-healing CFRPs can be roughly divided into two categories: intrinsic [9,10,11] and extrinsic [12,13,14] types. Intrinsic self-healing CFRPs are based on the inherent self-healing function of their polymer matrices (via reversible chemical bonds), which shows an advantage of high crack self-healing efficiency [8,15] but with the disadvantages of a complicated synthesis process and low mechanical properties [16]. However, extrinsic self-healing CFRPs achieve their crack self-healing by embedding microcapsules or micro-vessels (injected with healing agents, which flow out of broken capsules/vessels to repair cracks) or thermoplastics (the melt flows to fill and repair cracks when heated) into the polymer matrix of mature CFRPs [15]. Due to the ease of processing, a relatively low melting point (*T*_m_), and good melt flowability, the incorporation of thermoplastic healing agents into CFRP matrices is considered a promising approach to crack self-healing [17]. In addition, during thermoplastic melting and crack filling, hydrogen bonding or covalent bonding between the thermoplastic and the polymer matrix will increase the interfacial strength as well as the self-healing efficiency of the mechanical properties [18,19,20].

Previous studies have demonstrated that poly[ethylene-co-(methacrylic acid)] (EMAA) is an excellent self-healing thermoplastic copolymer with the advantages of a high thermal expansion coefficient, covalently crosslinking with epoxy, and pressure delivery self-healing mechanism [17,18,19,20,21]. Traditionally, EMAA has been incorporated into thermosetting epoxy matrices of composite laminates in the form of particles [18,19,20,22], membranes [23,24], meshes [25,26,27], stitches [28,29,30,31], or woven filaments [32,33,34,35], showing good delamination resistance and high recovery of interlaminar fracture toughness. Some pioneering studies carried out by Pingkarawat et al. [18,19,20,28,29,30] reported that EMAA can improve the critical strain energy release rate modes I (*G*_IC_) and II (*G*_IIC_) by 60–650% and 35–200%, respectively, with healing efficiencies of 40–300% for *G*_IC_ and 47–130% for *G*_IIC_. Similar explorations conducted by Varley et al. [14,26], Meure et al. [21,25], Yang et al. [31], Ladani et al. [32,33], and Loh et al. [34,35] have also found that increments of 13–400% for *G*_IC_ and 20–76% for *G*_IIC_, as well as healing efficiencies of 45–410% for *G*_IC_, 40–80% for *G*_IIC_, and 36–109% for interlaminar shear strength (*ILSS*) were respectively obtained. Nevertheless, the literature also revealed that the integration of less strong EMAA into composite laminates will decrease the in-plane strength (by 25–50%) [19,28] and *ILSS* (by 22–35%) [22,36], which is the biggest challenge for structural applications. In addition, another obvious disadvantage of previous studies is that the self-healing of cracks is achieved by heating the entire structure (usually using an oven).

In recent years, to address the above weaknesses, carbon nanotubes (CNTs) have been considered one of the most promising modified components of a self-healing agent due to their superior strength and stiffness and good electro-thermal performance. Especially due to the excellent Joule heating effect of CNTs and CNT-based composites [37,38,39,40], CNTs have been introduced into some self-healable polymers based on the Diels–Alder reaction [41,42,43], dynamic ionic bonds [44,45,46], supramolecular interactions [47,48,49], or covalent bonds [50]. These CNTs can form a continuous conductive network and can be regarded as an embedded nanoscopic heat resource when an electric current passes, activating the thermal self-healing capability of the self-mendable agent to heal cracks nearby. Additionally, a CNT/EMAA composite film was made by spraying in our prior study [17], and it showed good self-healing efficiency for mode III delamination cracks by using an in situ electrical heating method. However, how to combine EMAA and CNTs to obtain better healing efficiency for delaminated cracks has not been considered. It is still unknown how effective the CNT/EMAA composite film is for repairing mode I/II delamination cracks (tensile opening mode or in-plane shear mode of failure) by means of a localized electrical heating method.

Obviously, the presence of delamination cracks can lead to a significant reduction in the load-bearing capacity of the composite laminates and threaten the safety of structures. Therefore, in this work, we attempted to prepare two different sandwich and porous CNT/EMAA self-healing interleaves that could simultaneously increase the interfacial resistance of CFRPs in situ and self-heal delamination cracks via the electrical heating method. The two prepared sandwich and porous interleaves, CNT-EMAA-CNT and EMAA-CNT-EMAA, were respectively incorporated into the middle interfaces of CFRP laminates. Then, the toughening capability was evaluated by the double cantilever beam (DCB) and end-notched flexure (ENF) tests. The damaged specimens were repaired by electrical heating activation, and the corresponding crack self-healing efficiencies (η) were also assessed. A detailed microstructural analysis was carried out to determine the mechanisms of the interfacial toughening and electrical heating self-healing of delamination cracks.

## 2. Materials and Methods

### 2.1. Materials

EMAA (Model 426628, Sigma-Aldrich, St. Louis, MO, USA) was supplied in the form of particles with methacrylic acid (containing carboxyl groups: -COOH) at a weight ratio of 15%. Multi-walled CNTs were functionalized with amine groups (-NH2) with a weight fraction of 2.44%, an outer diameter of 8–15 nm, and a length of 30–50 μm. The NH2-CNTs, as a suspension in dimethylformamide solvent, were purchased from Chengdu Organic Chemicals Co. Ltd., where the NH2-CNT weight ratio was about 2.44%. The CFRP laminates in this work were prepared using a unidirectional carbon fiber fabric (Shenying Carbon Fiber co., Lianyungang, China) with an areal density of 300 g/m^2^ and a thickness of 0.177 mm, diglycidyl ether of bisphenol type-A epoxy (EP, Araldite^®^ LY1564, Huntsman, Woodlands, TX, USA), and a primary amine cure agent (DMDC, Aradur^®^ 2954, Huntsman, Woodlands, TX, USA).

### 2.2. Preparation

As shown in Figure 1a, porous CNT/EMAA membranes were prepared by hot pressing and spraying. Firstly, EMAA particles were uniformly distributed over a flat mold and hot-pressed to obtain a homogenous membrane with a thickness of 90 ± 6 μm, which was then hot-pressed using a template, as described in detail in our previous study [17]. Thus, porous EMAA films with a porosity of around 22.7% and a distance between the centers of two adjacent holes of 1.60 mm were prepared. After that, the CNT suspension was sprayed on the surface of the EMAA membranes using a spray gun. Some EMAA porous membranes were sprayed with CNTs on both sides (CNT-EMAA-CNT), with a CNT areal density of 5 g·m^−2^ on each side (a total CNT areal density of 10 g·m^−2^). Some other EMAA porous membranes were sprayed with CNTs just on one side (EMAA-CNT), with a CNT areal density of 10 g·m^−2^. Thereafter, nitric acid was coated on the surface of the sprayed CNTs for 75 s to remove the dispersant and then washed with deionized water. Finally, they were put into an oven and dried at 60 °C for 24 h (details can be seen in Appendix A). When the CNT-EMAA-CNT was incorporated into the CFRP, the porous CNT thin layers on the surface of EMAA absorbed the EP, and CNT/EP layers formed. Therefore, this in situ interleaf is named CNT/EP-EMAA-CNT/EP (Figure 1b). As for the EMAA-CNT, the EP was coated on the surface of the CNTs to form a CNT/EP layer, and then another EMAA film was laid on the CNT/EP. Thus, this interleaf is named EMAA-CNT/EP-EMAA (Figure 1c). Figure 1d shows the resistivity of the CNT/EP-EMAA-CNT/EP and EMAA-CNT/EP-EMAA interleaves. It can be seen that the resistivity decreased with the increased total CNT weight loading (*W*_CNT_). When the *W*_CNT_ reached about 10 g·m^−2^, the resistivity changed slowly or nearly steadily, indicating that the conductive paths formed by CNTs already reached a saturation state. Accordingly, Figure 1e,f shows the infrared temperature fields of the CNT/EP-EMAA-CNT/EP and EMAA-CNT/EP-EMAA interleaves, with a *W*_CNT_ of 10 g·m^−2^ at different input voltages during the electrical heating process. It can be seen that they both reached 150 °C (self-healing reaction temperature) within 4 min at the 32 V input voltage. Therefore, the CNT/EMAA, with a *W*_CNT_ of 10 g·m^−2^ was used to prepare both of the sandwich self-healing membranes. The preparation and characterization of the two “off-line” interleaves with various CNT weight loadings are described in detail in the Appendix A.

Then, the CNT/EP-EMAA-CNT/EP and EMAA-CNT/EP-EMAA interleaves were respectively incorporated into the middle interfaces of unidirectional CFRP laminates ([0°]_10_). For comparison, baseline CFRP laminate was also prepared. To measure the *G*_IC_ and *G*_IIC_ of the laminates, a 10 μm-thickness of Teflon film was inserted into one end of the middle interface of the laminates to produce a pre-crack. All laminates were fabricated by the hand lay-up and hot-pressing process, with an EP-to-DMDC weight ratio of 100:35. The laminates were first cured at 60 °C for 4 h and then post-cured at 150 °C for 2 h. During curing, the pressure was always 0.7 MPa. Then, the cured laminates with a fiber volume fraction of roughly 53% and a thickness of roughly 3.40 mm were obtained.

### 2.3. Characterization

To determine the healing temperature, differential scanning calorimetry (DSC, Q20, TA) of the EMAA and the cured epoxy system were performed to determine the *T*_m_ and the glass transition temperature (*T*_g_), with a heating rate of 10 °C/min. Fourier transform infrared (FTIR, Nicolet 6700 Spectrometer) spectroscopy of the EP system, EMAA, and CNT/EP/EMAA was used to identify the potential chemical reactions and related self-healing mechanisms. The specimens were characterized by a Nicolet 6700 Spectrometer FTIR, scanning in the range of 500–4000 cm^−1^ in the attenuated total reflection mode.

According to the ASTM D5528 standard, the *G*_IC_ was measured using DCB specimens, with dimensions of 150 mm × 20 mm and a pre-crack length of 50 mm, as shown in Figure 2a. The DCB samples were loaded at a speed of 2 mm/min, and the testing was terminated as soon as the pre-crack tip extended forward by approximately 5 mm. Then, the *G*_IC_ was calculated using the formula *G*_IC_ = 3*Pδ*/2*b*(*a* + |Δ|), where *b* is the width of the specimens and Δ is a correction factor discovered empirically from the plot of *C*^1/3^ (specimen compliance) versus the delamination length. *P* and *δ*, respectively, stand for the maximum load and corresponding opening displacement, and *a* = 50 + Δ*a* (Δ*a* is the crack propagation length) represents the total crack length corresponding to the *P*.

According to the AITM 1.0006 standard of Airbus, the *G*_IIC_ was measured using ENF specimens in a 3-point bending test, with dimensions of 150 mm × 20 mm, as shown in Figure 2b, where the span length (2*L*) and pre-crack length were respectively set as 100 and 25 mm. The ENF test specimens were loaded at a displacement rate of 1 mm/min, and five specimens were tested for each laminate. The *G*_IIC_ was calculated according to *G*_IIC_ = 9*Pδa^2^*/2*b*(2*L^3^* + 3*a*^3^), in which, *b* is the width of the samples, *a* is the total delamination length (i.e., *a* = 25 + Δ*a*), and *P* and *δ* represent the load and corresponding bending displacement, respectively. For brittle materials, the moment of unstable crack growth can be clearly determined, and *P*_max_ is attributed to the critical load at the onset of crack growth. However, for hybrid materials containing a ductile interleave layer through which the crack is growing, the crack initiation occurs long before *P*_max_. [1,3,13] Thus, crack propagation was monitored using a high-resolution digital camera system, and the *G*_IIC_ and the self-healing efficiency were evaluated based on the visual crack initiation point (VIS) for the studied laminates.

The delaminated specimens were clamped to make the fractured surfaces connect with each other, followed by electrical heating to 150 °C for 1 h to heal the crack in situ, and then cooled down to room temperature. The applied currents for the CNT/EP-EMAA-CNT/EP and EMAA-CNT/EP-EMAA interleaved specimens were about 15 V and 18 V, respectively. Two silver electrodes were set on the upper and bottom surfaces of the specimens and connected with copper wires and the power supply (MP3020D, MAISHENG), as shown in Figure 2. A FLIR ONE infrared camera and a HJ-500x desktop magnifier were employed to record the temperature fields and related crack healing evolvement of the specimens during the electrical heating process. The healed specimens were tested again, then healed and tested once again. Thus, the first (*η*_1_) and second (*η*_2_) self-healing efficiencies of the *G*_IC_ and *G*_IIC_ can be evaluated. The *η* was calculated according to the *η* = *G*_healed_/*G*_original_, where the *G*_original_ and *G*_healed_ are the original and the self-healed values.

Field emission scanning electron microscopy (FESEM, MIRA3 TESCAN, Czech Republic) was conducted to evaluate the geometry, the distribution of the self-healing agent, and the fracture surfaces of the various interleaves used. Simultaneously, optical microscopy (50 ×, SD-Y2001) was used to observe the delamination propagation behavior of the CFRP laminates.

## 3. Results and Discussion

### 3.1. Self-Healing Mechanism

Figure 3a presents the DSC curves of the EMAA and cured epoxy system. It is obvious that the *T*_m_ of EMAA used in this study was 70.4 °C, which was determined by the second endothermic peak. In addition, the first endothermic peak of the EMAA denotes a phase transition due to its purity. Furthermore, the DSC curve of the cured epoxy resin indicates a glass transition temperature (*T*_g_) of around 140 °C, which was prepared according to the curing regime described in Section 2.2. This *T*_g_ is lower than the temperature at which the condensation reaction takes place at the healing stage (150 °C) [18,24], contributing to the enhancement of the efficiency of the healing reaction.

Figure 3b shows the FTIR spectra of the EMAA, EP system, and CNT/EP/EMAA, which were determined to trace the chemical crosslinking between thermoplastic EMAA and thermoset EP. It reveals that the characteristic peaks at 1696 cm^−1^ and 936 cm^−1^ correspond to the C=O stretching of the carboxyl groups in the EMAA. As for the CNT/EP/EMAA system treated by pre-curing at 60 °C for 4 h, the same peaks are relatively weaker than that of the pure EMAA, because of the hydrogen or ionic bonding between the EMAA and CNT/EP (Equations (1) and (2) in Figure 3c) [21]. Undoubtedly, these non-covalent bond crosslinks help to strengthen the interfacial binding between the EMAA and CNT/EP. The characteristic peak at 1581 cm^−1^ represents the amino group in the EP system or NH_2_-CNT. In addition, the characteristic peak of the epoxy group in EP at 915 cm^−1^ indicates that the epoxy group in this system was not completely consumed after pre-curing. When the CNT/EP/EMAA system was further treated at 150 °C for 2 h, the characteristic peak at 1696 cm^−1^ decreased but an ester peak at 1732 cm^−1^ appeared. This means that the carboxyl group in the EMAA was condensed with the epoxy group or hydroxyl group in the EP matrix to form the ester group, as can be seen in the chemical Equations (3) and (4) in Figure 3c [21,22]. In addition, Figure 3b indicates that after curing, all epoxy groups were consumed because of the disappearance of the corresponding transmittance peek at 915 cm^−1^. In conclusion, these dense crosslinks can help to restore the mechanical properties of the healed system.

### 3.2. Microstructures

Figure 4 shows the side sections of the laminates, respectively interleaved with CNT/EP-EMAA-CNT/EP and EMAA-CNT/EP-EMAA. As can be seen in Figure 4a,b, the CNT/EP-EMAA-CNT/EP interleaf was a sandwich structure with CNT/EP face layers (~8.5 μm, Figure 4c), a localized EMAA core (~95 μm, Figure 4b), and periodic EP column arrays. Figure 4c reveals that the CNT/EP layer with relatively stable thickness was well combined with the adjacent EMAA or CF layers, and the CNTs were evenly distributed in the EP matrix (subfigure in Figure 4c). As for the EMAA-CNT/EP-EMAA interleaf, a sandwich structure consisting of the localized EMAA face layers and CNT/EP core can be seen in Figure 4d,e. It can be seen that the interleaf has a total thickness of approximately 185 μm and the EMAA layers exhibited good interfacial bonding with the surrounding CF layers. Similarly, the CNT layer in the EMAA-CNT/EP-EMAA interleaf was thoroughly infiltrated by EP with consistent thickness (about 18 μm), and the CNTs, EP, and EMAA were tightly crosslinked at the interface, as can be seen in Figure 4f (zoomed-in view of Figure 4e).

### 3.3. Toughening and Self-Healing of G_IC_

Figure 5a–c displays some typical load–displacement curves of the baseline CNT/EP-EMAA-CNT/EP and EMAA-CNT/EP-EMAA CFRP from the DCB testing before and after healings. The peak loads and accompanying displacements on the original curves (marked in blue) of the CNT/EP-EMAA-CNT/EP and EMAA-CNT/EP-EMAA interleaved CFRP were larger than those of the baseline CFRP. The average *G*_IC_ values from the toughening laminates were calculated as 0.40 ± 0.04 and 0.54 ± 0.06 kJ·m^−2^ for improvements of 24.0% and 66.9%, respectively, when compared to the baseline CFRP (0.33 ± 0.01 kJ·m^−2^), as shown in Figure 5d (in blue). The improvement in the *G*_IC_ was due to the fact that the EMAA formed a large-scale extrinsic bridging zone along the delamination crack, as shown in the subfigures of Figure 5b,c. Because of the high ductility and excellent adhesion ability, the bridged EMAA filaments underwent large normal deformation and generated a mode I traction load that was opposite to the crack opening, which reduced the stress applied at the crack tip, thereby increasing the *G*_IC_ [28].

As shown in Figure 5e, the failure modes of the base CFRP are mainly the breakages of the EP matrix and bridging carbon fibers, as well as the debonding of the fiber/matrix interface. However, some plastic deformation zones are formed on the fractured surfaces of the CNT/EP-EMAA-CNT/EP (Figure 5(f_1_)) and EMAA-CNT/EP-EMAA (Figure 5(g_1_,g_2_)) interleaved laminates, where some ductile tearing of the EMAA can be observed. This is the reason for the enhancement of fracture toughness. In addition, compared to the CNT/EP-EMAA-CNT/EP interleaf, more bridging EMAA filaments (subfigures of Figure 5b,c) in the EMAA-CNT/EP-EMAA interleaf occurred, which could also bond with the CF layers, thus showing a better toughening effect. Moreover, some micro-voids were formed within the EMAA zones (Figure 5(f_2_,g_3_)), which were induced by the esterification reaction between the EMAA and the EP.

The damaged specimens after the original tests were healed in situ by the electrical heating-activated method, and the corresponding healing process of the EMAA-CNT/EP-EMAA interleaved CFRP is shown in Figure 6. By comparing Figure 6a,b, it can be found that the EMAA near the original delamination crack melted after 15 min of electrical heating at 150 °C, and the crack zones filled with EMAA became dark (marked with pink arrows). Figure 6c,d reveals that the crack region was gradually filled by the melted EMAA after continuous electric heating for 30 min and 45 min, increasing the amount of dark EMAA there. The corresponding temperature fields of the side and top views are shown in Figure 6(c’,d’), indicating that the healing temperature remained almost constant at 150 °C. The healing process was essentially completed within 60 min. The side section of the specimen after the first healing was polished (Figure 6e), which indicates that the delamination crack was repaired when compared to the damaged one (Figure 6a). The healed specimens were loaded again and second delaminated cracks occurred (Figure 6f), which could be healed once again (Figure 6g) by the same electrical heating-activated method.

Figure 5b,c presents the load–displacement curves of the CNT/EP-EMAA-CNT/EP and EMAA-CNT/EP-EMAA specimens after the first and second healings, respectively. The results show that the peak loads were almost completely recovered, indicating that the delamination damage could be repaired. However, with an increasing number of repairs, the slopes of the load–displacement curves marginally declined, indicating that the stiffness of the DCB specimens could not be fully recovered. This is because during the self-healing process, molten EMAA flowed into the delamination crack regions, and an increase in the EMAA concentration enhanced the toughness and reduced the stiffness of these regions accordingly. As for the CNT/EP-EMAA-CNT/EP-interleaved CFRP, the obtained *G*_IC_ after the first and second healings were respectively obtained as 0.44 ± 0.02 and 0.50 ± 0.04 kJ·m^−2^, accompanied by a *η*_1_ of 109.7% and a *η*_2_ of 123.5%, when compared to its original value (0.40 ± 0.04 kJ·m^−2^). Likewise, the *G*_IC_ of the EMAA-CNT/EP-EMAA-interleaved CFRP after the first and second healings were respectively measured as 0.67 ± 0.05 and 0.68 ± 0.04 kJ·m^−2^, accompanied by a *η*_1_ of 122.7% and a *η*_2_ of 125.9%, when compared to its original value (0.54 ± 0.06 kJ·m^−2^). Thus, the results prove that the *G*_IC_ of these three interleaved laminates can be restored completely. In addition, the healing efficiency increased slightly with the increase in healing times, and this is because the EMAA content of the crack regions increased more and more with the increased healing times.

As can be seen in Figure 5(f_2_,g_3_), micro-voids within the EMAA phase were formed by the produced bubbles from the esterification reaction between the -COOH groups in the EMAA and the hydroxyl groups in the EP, which provided the pressure-driven delivery mechanism during the healing process [21,22,28]. The bubbles helped the molten EMAA to flow into the delamination crack under internal pressure and then bound with the fractured surfaces. In addition, the -COOH groups in EMAA can also react with the -NH_2_ groups in CNTs [22,34], which further increases the crosslinking density between different components, resulting in more energy consumption of the extracted NH_2_-CNTs during crack growth (subfigures in Figure 5(f_2_,g_3_)). In conclusion, all these factors together led to an improvement in the delamination resistance and high self-healing efficiency.

### 3.4. Toughening and Self-Healing of G_IIC_

Figure 7a–c shows the load–displacement curves of the studied laminates before and after the healings, and the obtained *G*_IIC_ is shown in Figure 7d. For the baseline CFRP, the applied load increased almost linearly until the onset of the pre-crack propagation, where the load dropped suddenly, suggesting a typical unstable propagation of mode II cracking. Meanwhile, the interleaved laminates show a distinguishable nonlinear load–displacement relationship, and no significant load drop appeared when the delamination cracks began to propagate. This means that the delamination cracks propagated steadily. At the visible onset of the delamination cracks, as seen in Figure 7b,c, the extended crack length was about 3–5 mm (marked with black stars). The average *G*_IIC_ values of the CNT/EP-EMAA-CNT/EP and EMAA-CNT/EP-EMAA interleaved laminates were calculated as 0.83 ± 0.01 and 0.84 ± 0.04 kJ·m^−2^, respectively, exhibiting increments of 15.3% and 16.7%, respectively, when compared to that of the baseline CFRP (0.72 ± 0.04 kJ·m^−2^), as shown in Figure 7d. Under the mode II loading, the middle interface was mainly affected by the interlaminar shear stress, but the principal tension stress was generated at the 45° direction to the delamination plane, which resulted in tension microcracks at the middle interface (45° to the shear load direction). As the load increased, these microcracks tended to form S-type ligaments and then merged at the upper/lower boundary of the middle interface, enabling macroscopic delamination crack propagation [1]. Due to the lower strength/stiffness of EMAA [17], the maximum normal stress that EMAA can withstand is lower than that of the EP matrix, which may be the main reason for the decreased *G*_IIC_ of the EMAA-interleaved CFRP.

Figure 8 displays the FESEM images of the fractured side sections and surfaces of the ENF specimens. The delamination crack of the baseline CFRP propagated straightly (see Figure 8(a_1_)), which was caused by the unsteady propagation of cracking. A zoomed-in picture of the delamination crack zone is shown in the subfigure of Figure 8(a_1_), showing debonding between the carbon fibers (with a smooth surface) and the EP, as well as the creation and merging of characteristic microcracks during the mode II fracture. This is further proved by the fractured surfaces shown in Figure 8(a_2_,a_3_), where debonding between the carbon fibers and the EP and the brittle fracture of the EP are fairly obvious. As for the CNT/EP-EMAA-CNT/EP- and EMAA-CNT/EP-EMAA-interleaved laminates, as shown in Figure 8(b_1_–b_3_,c_1_–c_3_), the delamination cracks are somewhat zig-zagged, and they mainly propagated within the interleaves (Figure 8(b_1_,c_1_)). This result suggests that the interface between the interleaf and the adjacent CF layers was more resistant to delamination than the interleaf itself, thus causing cracks to propagate along lower-energy paths inside the interleaf. Meanwhile, some extracted CNTs (subfigures of Figure 8(b_1_,b_3_,c_1_,c_3_)) further reveal that more energy could be absorbed during the delamination, which would help to improve the *G*_IIC_ of the CNT/EP-EMAA-CNT/EP- and EMAA-CNT/EP-EMAA-interleaved laminates.

The original delaminated specimens were healed by the electrical heating method and then reloaded, followed by re-healing and re-loading once again. The obtained load–displacement curves from the first and second healings are also shown in Figure 7b,c. All curves show similar trends, i.e., the initial linear and late nonlinear load–displacement relationships, as well as slightly reduced slopes of the curves. The healed first and second *G*_IIC_ of the CNT/EP-EMAA-CNT/EP- and EMAA-CNT/EP-EMAA-interleaved laminates were respectively averaged as 0.76 ± 0.03 and 0.75 ± 0.03 kJ·m^−2^, 0.80 ± 0.04 and 0.78 ± 0.04 kJ·m^−2^ (see Figure 7d), suggesting that the *η*_1_ and *η*_2_ of the *G*_IIC_ were obtained as 91.2% and 90.6% for the CNT/EP-EMAA-CNT/EP-interleaved laminate, and 94.7% and 93.1% for the EMAA-CNT/EP-EMAA-interleaved laminate, respectively. Therefore, it can be seen that the recovered *G*_IIC_ of the laminates with the CNT-reinforced EMAA interleaves was still greater than that of the baseline CFRP. However, the healed *G*_IIC_ could not be restored to its original value, and this is because the EMAA content and micro-voids (see Figure 8(b_3_,c_3_)) in the crack zone increased with the healing times, and the principal tension stress (at the 45° direction to the delamination) that the healed crack zone could withstand was reduced accordingly.

## 4. Conclusions

This paper presents the preparation of two different sandwich CNT/EMAA self-healing interleaves that can act as a crack self-healing interface activated by the in situ electrical heating of the CNTs. Firstly, their electrothermal performance and self-healing mechanisms were studied. It was found that when the total CNT areal density was 10 g·m^−2^, the temperature of both the CNT/EP-EMAA-CNT/EP and EMAA-CNT/EP-EMAA sandwich interleaves could quickly increase to 150 °C (required for the esterification reaction during the crack healing process) by electrical heating, showing an excellent electrical heating capability. Secondly, their capability for toughening and self-healing the CFRPs was evaluated. The results show that both the CNT/EP-EMAA-CNT/EP and EMAA-CNT/EP-EMAA sandwich interleaves could improve the *G*_IC_ by 24.0% and 66.9%, and the *G*_IIC_ by 15.2% and 16.7%, respectively. Simultaneously, the CNT/EP-EMAA-CNT/EP interleaf exhibited self-healing efficiencies of 109.7–123.5% for the *G*_IC_ and 90.6–91.2% for the *G*_IIC_, respectively. The EMAA-CNT/EP-EMAA interleaf exhibited self-healing efficiencies of 122.7–125.9% for the *G*_IC_ and 93.1–94.7% for the *G*_IIC_, respectively. In conclusion, the proposed EMAA-CNT/EP-EMAA interleaf is preferable when used as a crack healing agent via the electrical heating method, with a high healing efficiency and reasonable toughening capability.

## Figures and Tables

**Figure 1 polymers-14-04313-f001:**
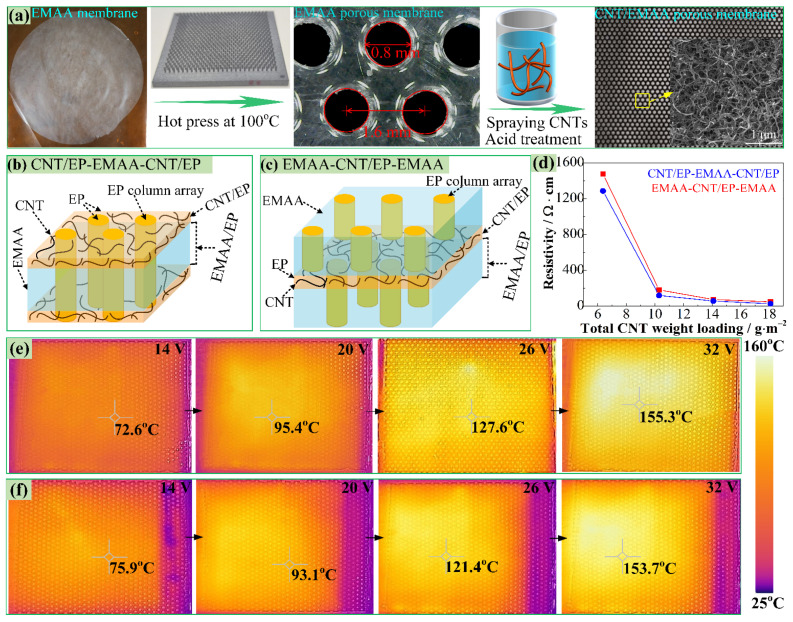
(**a**) Fabrication of CNT/EMAA porous membrane, (**b**,**c**) schematic diagrams of CNT/EP-EMAA-CNT/EP and EMAA-CNT/EP-EMAA interleaves, (**d**) resistivity of the interleaves versus the total CNT weight loadings, (**e**,**f**) infrared temperature fields of CNT/EP-EMAA-CNT/EP and EMAA-CNT/EP-EMAA at different input voltages.

**Figure 2 polymers-14-04313-f002:**
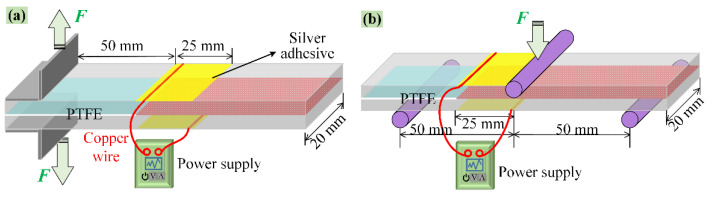
(**a**) DCB specimen and crack healing by electrical heating, (**b**) ENF specimen and crack healing by electrical heating.

**Figure 3 polymers-14-04313-f003:**
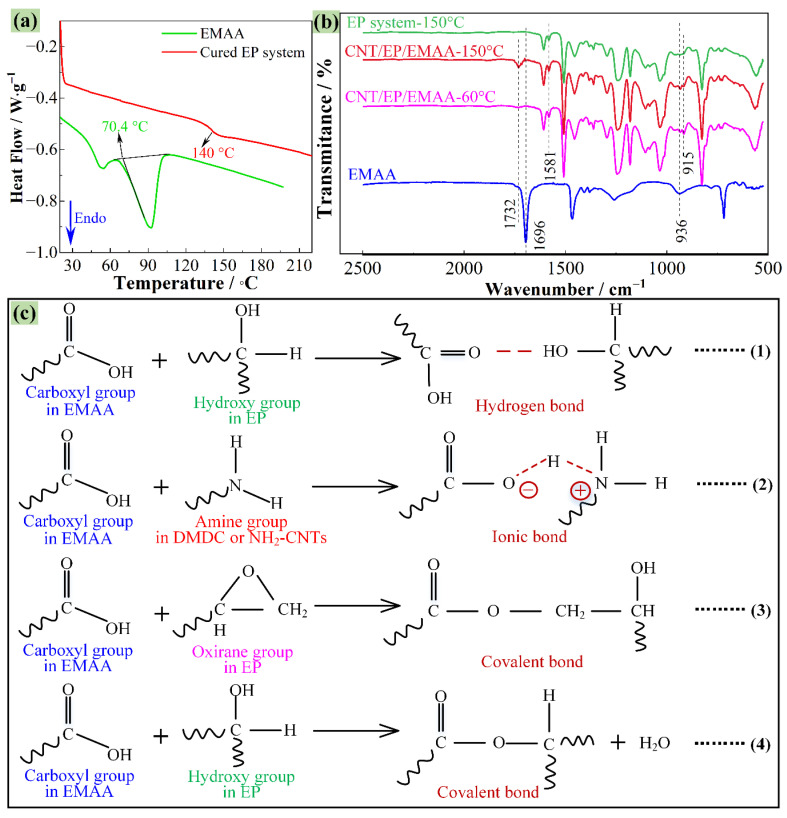
(**a**) DSC curves of EMAA and cured epoxy system, (**b**) FTIR spectra of EMAA, EP, and CNT/EP/EMAA system, (**c**) potential reactions between EMAA and epoxy and harder groups and CNTs.

**Figure 4 polymers-14-04313-f004:**
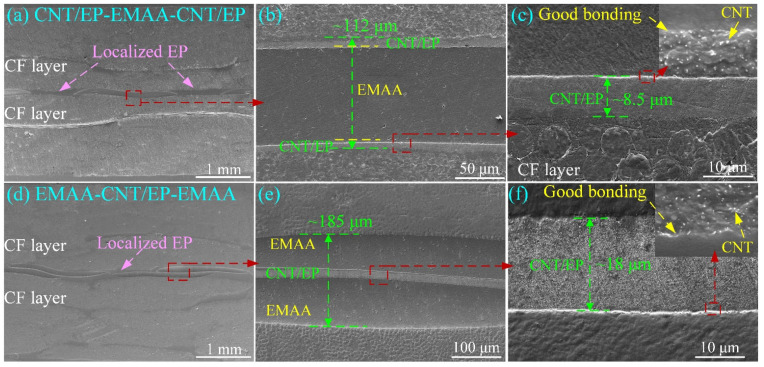
FESEM images of cured laminates, (**a**–**c**) CFRP with the CNT/EP-EMAA-CNT/EP interleaf, (**d**–**f**) CFRP with the EMAA-CNT/EP-EMAA interleaf.

**Figure 5 polymers-14-04313-f005:**
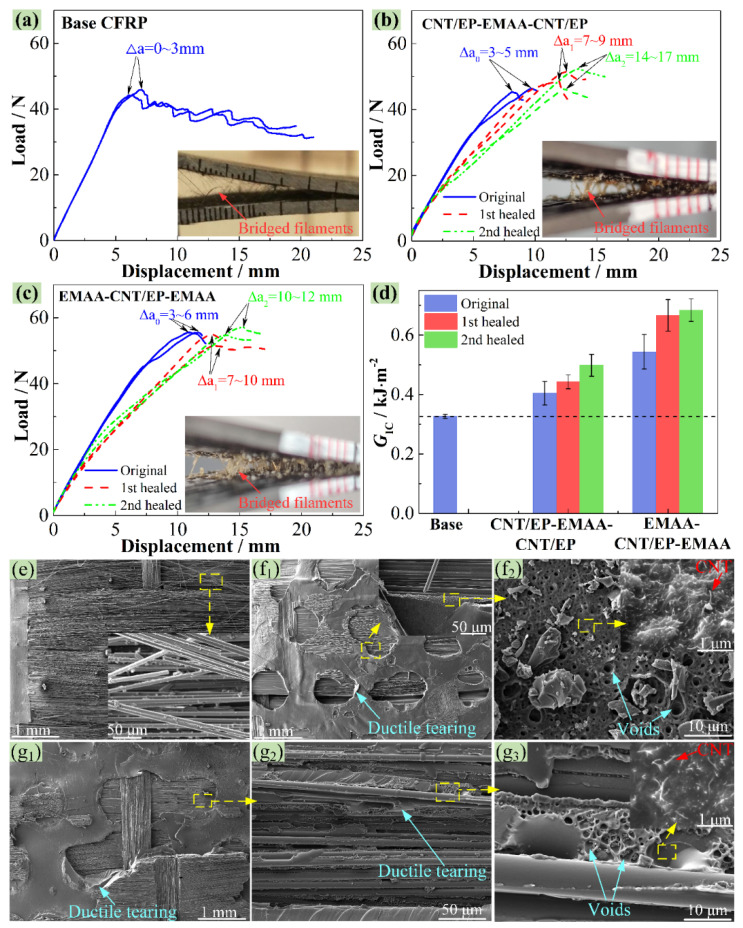
DCB testing results before and after self-healings, load–displacement curves of (**a**) base CFRP, (**b**) CNT/EP-EMAA-CNT/EP CFRP and (**c**) EMAA-CNT/EP-EMAA CFRP, (**d**) comparison of *G*_IC_, SEM images of the fractured DCB specimens, (**e**) base CFRP, (**f_1_**,**f_2_**) CNT/EP-EMAA-CNT/EP CFRP and (**g_1_**–**g_3_**) EMAA-CNT/EP-EMAA CFRP.

**Figure 6 polymers-14-04313-f006:**
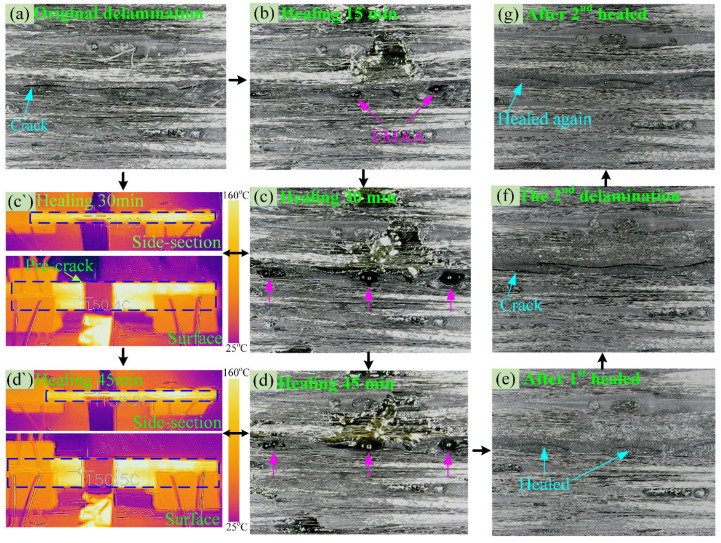
(**a**–**g**) Optical pictures of the crack from two delamination-healing cycles; (**c`**,**d`**) corresponding infrared temperature fields after healing 30 and 45 min by electrical heating.

**Figure 7 polymers-14-04313-f007:**
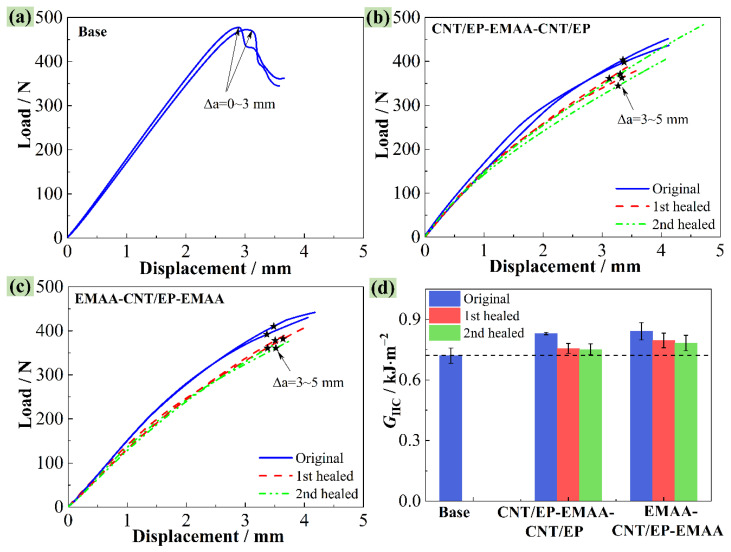
Mode II load–displacement curves from two delamination–healing cycles, (**a**) baseline CFRP, (**b**) CNT/EP-EMAA-CNT/EP CFRP, (**c**) EMAA-CNT/EP-EMAA CFRP, (**d**) comparison of *G*_IIC_.

**Figure 8 polymers-14-04313-f008:**
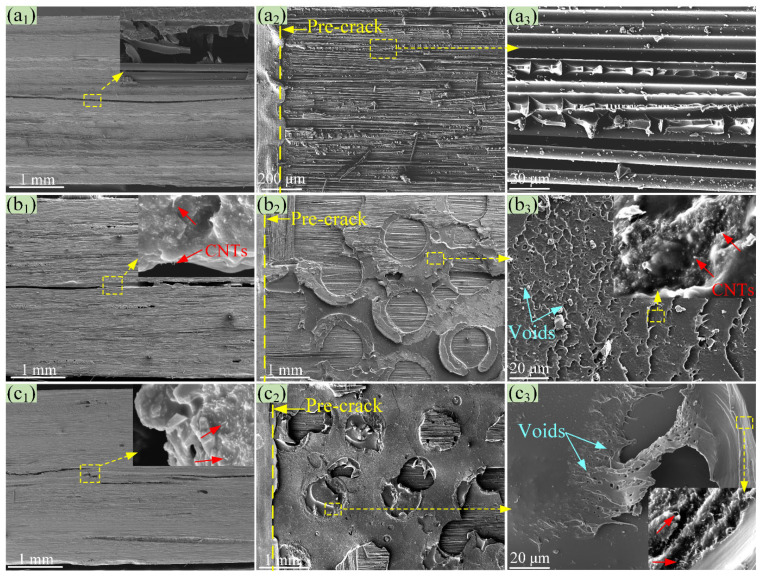
SEM images of the fractured ENF specimens, (**a_1_**–**a_3_**) baseline CFRP, (**b_1_**–**b_3_**) CNT/EP-EMAA-CNT/EP CFRP, (**c_1_**–**c_3_**) EMAA-CNT/EP-EMAA CFRP.

## Data Availability

The data presented in this study are available upon request from the corresponding author.

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
