# Peer review of "Electrothermally Self-Healing Delamination Cracks in Carbon/Epoxy Composites Using Sandwich and Tough Carbon Nanotube/Copolymer Interleaves"

_polymers, 2022, doi:10.3390/polym14204313_

Round 1

Reviewer 1 Report

This manuscript describes the preparation of CNT/EP-EMAA-CNT/EP and EMAA-CNT/EP-EMAA sandwich interleaves introduced into unidirection CFRP laminates which can toughen and electrothermally self-heal delamination cracks. Authors completely investigated the areal density of CNTs (electro-thermal self-healing agent) and the temperature increasing when input different voltages in interleaves to clarify its electrical-healing capability. Furthermore, mode I (GIC) and II (GIIC) critical strain energy release and the respective self-healing efficiency was systematically measured. Due to the increasing EMAA content in the crack zones with the healing time, the GIC value rose after each self-healing but the opposite happened with the GIIC value. In my opinion, this manuscript need to have minor revission to be published in this journal, following reasons:

1.     At paragraph 2.2., please explain more about acid treatment to remove dispersant which did not clarify in the citation [17].

2.     The confusion can be seen at Figure 3b and its explanation. The peak 936cm-1 of the CNT/EP/EMAA specimen treated 150oC for 2h did not become weaker (as shown in Figure 3b).

3.     At paragraph 3.4., there have an argument that “The average GIIC values of CNT/EP-EMAA-CNT/EP and EMAA-CNT/EP-EMAA interleaved laminates are calculated as 0.83±0.01 and 0.84±0.04 KJ.m-2, respectively, exhibiting a decrement or increment of 15.2% and 16.7% when respectively compared with that of the base CFRP (0.72±0.04 KJ.m-2). While the two original sandwich interleave columns in Figure 7d were both higher than CFRP base one. Please review and explain.

In my opinion, this manuscript should be submitted after minor revision. Therefore, I recommend accepting this manuscript for publication in this journal.

Reviewer 2 Report

The article "Electrothermally Self-healing Delamination Cracks in Carbon/Epoxy Composites Using Sandwich and Tough Carbon nanotube/Copolymer Interleaves" describes the design of a self-healing composite (carbon fiber reinforced plastic). It is a valuable study that can be published after authors address the following problems:

The English language needs some polishing for style and typos (e.g. use DSC curve instead of thermogram, use kJ instead of KJ).

Please indicate in DSC figure the direction of exo or endo with an arrow. Also the Oy axe is missing together with the values. If the scale for the DSC curves EMAA and cured epoxy system are not compatible, please use left-right Oy axes.

Reviewer 3 Report

The authors submitted a manuscript entitled “Electrothermally Self-healing Delamination Cracks in Carbon/Epoxy Composites Using Sandwich and Tough Carbon nanotube/Copolymer Interleaves” with the reference polymers-1955605.

The subject of the manuscript is really interesting. The authors prove their idea to work. There still are some issues to clarify if there are more than three layers in the composite or how to heat just a part of the surface (especially when the nanotubes layer is not in the outside). Maybe these are issues for future work.

The paper is well designed, the English is good, and the measurements, tests and ideas are well explained and also look experimentally carefully done. Overall, the manuscript is ok, the results seem sound, the graphical part is clear (see some suggestions to improve this part), the references well-presented and adequate to the Polymers magazine.

Probably the caption of the figures could be a little bit more descriptive in order to allow the reader to understand the entire figure without reading the full text.

The supplementary material is so small, compared with the manuscript, that it will be a better solution to include it in the manuscript.

The paper deserves to be published with some minor corrections:

1 – Line 54, However in the beginning of the sentence does not make sense in this case. Please cut it.

2 – Line 68; literature instead of literatures.

3 – Line 86. Despite being known that Mode I refers to opening and Mode II refers to in-plane shear movement, since the journal will have readers that are not experts on the subject it will be an improvement to detail this a little bit in the text.

4 – Line 232 and Figure 3(b). The peak in the text is at 1731 cm-1 and in the figure at 1732 cm-1. Please, correct text.

5 – Line 261. kJ is written with a minuscule k (not KJ).
